

# The detailed distribution of T cell subpopulations in immune-stable renal allograft recipients: a single center study

Quan Zhuang*, Bo Peng*, Wei Wei, Hang Gong, Meng Yu, Min Yang, Lian Liu and Yingzi Ming

Transplantation Center, The 3rd Xiangya Hospital of Central South University, Changsha, Hunan, China
* These authors contributed equally to this work.

## ABSTRACT

**Background:** Most renal allograft recipients reach a stable immune state (neither rejection nor infection) after transplantation. However, the detailed distribution of overall T lymphocyte subsets in the peripheral blood of these immune-stable renal transplant recipients remains unclear. We aim to identify differences between this stable immune state and a healthy immune state.

**Methods:** In total, 103 recipients underwent renal transplantation from 2012 to 2016 and received regular follow-up in our clinic. A total of 88 of these 103 recipients were enrolled in our study according to the inclusion and exclusion criteria. A total of 47 patients were 1 year post-transplantation, and 41 were 5 years post-transplantation. In addition, 41 healthy volunteers were recruited from our physical examination clinic. Detailed T cell subpopulations from the peripheral blood were assessed via flow cytometry. The parental frequency of each subset was calculated and compared among the diverse groups.

**Results:** The demographics and baseline characteristics of every group were analyzed. The frequency of total T cells (CD3+) was decreased in the renal allograft recipients. No difference in the variation of the CD4+, CD8+, and activated (HLA-DR+) T cell subsets was noted among the diverse groups. Regarding T cell receptor (TCR) markers, significant reductions were found in the proportion of γδ T cells and their Vδ2 subset in the renal allograft recipients. The proportions of both CD4+ and CD8+ programmed cell death protein (PD) 1+ T cell subsets were increased in the renal allograft recipients. The CD27+CD28+ T cell proportions in both the CD4+ and CD8+ populations were significantly decreased in the allograft recipients, but the opposite results were found for both CD4+ and CD8+ CD27-CD28- T cells. An increased percentage of CD4+ effector memory T cells and a declined fraction of CD8+ central memory T cells were found in the renal allograft recipients.

**Conclusion:** Limited differences in general T cell subsets (CD4+, CD8+, and HLA-DR+) were noted. However, obvious differences between renal allograft recipients and healthy volunteers were identified with TCR, PD1, costimulatory molecules, and memory T cell markers.

Corresponding author
Yingzi Ming,
myz_china@aliyun.com

## INTRODUCTION

Renal transplantation is still the most applicable management strategy for end-stage renal disease (ESRD) (*Darres et al., 2018*). Every patient needs to take immunosuppressants (ISs) to prevent rejection after organ transplantation, but taking an overdose of ISs will cause infection and drug toxicity (*Mota et al., 2013*). After a period of continuous monitoring and adjustment of the ISs concentration and immunocompetency, renal allograft recipients can achieve a stable immune state (neither infection nor rejection). However, this stable immune state is complex and influenced by multiple factors, such as ISs, allograft immune activity and psychological changes. Therefore, this immune-stable state should be different from healthy immunity.

Renal allograft recipients mainly take calcineurin inhibitors (CNIs) to prevent cellular rejection responses after transplantation (*Yu et al., 2018*). The most commonly applied CNIs in our department is tacrolimus (also known as FK506). The immunosuppressive activities of tacrolimus and other commonly used ISs (e.g., mycophenolate mofetil (MMF) and steroids) mainly impact T lymphocytes (*Hartono, Muthukumar & Suthanthiran, 2013*). Therefore, the constitution and proportion of T cells and their subpopulations should receive considerable attention. The chief subpopulations of T cells are indicated by the cell surface markers CD4 and CD8 together with CC chemokine receptor 7 (CCR7) and CD45RA (*Busch et al., 2016*). The activated subpopulations of each of these cell types can be defined by the supplement of activation markers, such as CD38 and HLA-DR (*Ferreira, Kumar & Humar, 2018*; *Tanko et al., 2018*). The naive T cell, effector memory (EM), central memory (CM), and effector T cell subsets were first defined based on CCR7 and CD45RA expression (*Sallusto et al., 1999*). The specific mechanism for the above ISs is inhibition of the production of interleukin-2, which allows naïve T cells to differentiate into effector T cells in lymphatic tissues (*Hartono, Muthukumar & Suthanthiran, 2013*). Compared with the activation process in naïve T cells, memory T cells are activated by antigen-presenting cells (APC) in diverse tissues, including organ allografts and are less susceptible to suppression by immunosuppressive agents and tolerogenic cells (*Yang et al., 2007*), whereas allo-reactive CD8+ memory T cells are known to be more resistant than CD4+ T cells (*Perez-Gutierrez et al., 2018*). Both CM and EM T cells show potential for generation following acute allograft rejection (*Danger, Sawitzki & Brouard, 2016*; *Siu et al., 2018*). Additionally, T cells need two signals for activation: the T cell receptor (TCR) and costimulatory molecules (*Agarwal & Newell, 2008*). There are two major categories of costimulatory molecules: the immunoglobulin superfamily (i.e., CD28) and the tumor necrosis factor receptor superfamily (i.e., CD27) (*Croft, 2003*). More recently, T cell immunosenescence has been shown to related to telomere-dependent replication senescence, and its characteristic phenotype and functional spectrum are associated with CD57 expression (*Cura Daball et al., 2018*), which is also associated with altered function (*Larbi & Fulop, 2014*). These CD57+ T cells were initially defined as being unable to proliferate under antigen stimulation, but recent studies showed that they could enter the active cell cycle and proliferate under certain stimulation conditions and

**Table 1 Inclusion and exclusion criteria in our study.**

| Inclusion criteria | Exclusion criteria |
|---|---|
| CDC test negative on the day of transplant | Existed DSA on the day of transplant |
| WBC count, total platelet count, and renal function within normal limits | Experienced proved bacterial and fungi infection at time of transplantation and (or) blood collection |
| Took FK506+MMF+Pred for their long-term maintenance immunosuppressant | Undertook organ transplantation previously |
|  | Had a history of malignancy |
|  | Serum positive for HIV, HCV antibody, or HBsAg (the latest result before transplantation) |
|  | Pregnancy at time of blood collection |
|  | Received a lymphocyte depleting therapy |

Note:
CDC, complement-dependent cytotoxicity; WBC, white blood cell; Pred, prednisolone; DSA, donor specific antibody; HIV, human immunodeficiency virus; HCV, hepatitis C virus; HBsAg, hepatitis B surface antigen.

maintain cytokine production (*Strioga, Pasukoniene & Characiejus, 2011*). Several studies clearly demonstrated that CD8+CD57+ T cells showed immunosuppressive activity and were more active in ISs-treated and human immunodeficiency virus-infected patients (*Frassanito et al., 1998*; *Sadat-Sowti et al., 1994*). The programmed cell death protein 1 (PD1) plays a role in chronic infection and organ transplant tolerance, and PD1 upregulation is associated with T cell exhaustion phenotypes in multiple animal models (*Wang, Han & Hancock, 2007*).

Continuous monitoring of immune cells is important for disease treatment and prognostic prediction (*Goldschmidt et al., 2018*). The peripheral blood is very important for assessing the immune state, because it is convenient to be collected and contains abundant significant information concerning the arrival of immune cells in the cognate tissue through the circulation (*Ruhle et al., 2016*). Multicolor flow cytometry is considered as the preferred scheme for analysis of blood samples, because it provides highly specific single cell levels with various indexes and high output characteristics (*Streitz et al., 2013*).

In our study, we obtained peripheral blood specimens from recipients 1 and 5 years post-kidney transplantation and analyzed their detailed T cell subset immunophenotyping using multicolor flow cytometry. We compared these profiles with those of healthy volunteers to assess the constitution and frequency of T cell subpopulations in immune-stable recipients.

## METHODS AND MATERIALS

### Study population and blood specimen collection

In total, 103 recipients aged 18–65 years old underwent kidney transplantation from January 1, 2012 to December 31, 2016 and received regular follow-up in the clinic of the 3rd Xiangya Hospital, Central South University. A total of 88 of these 103 recipients were eligible to be collected as cases for further investigation according to the key inclusion and exclusion criteria (Table 1). The recipients were allocated into two groups dependent

**Table 2 Overview of the two staining panels each dedicated to a specific cell type which is indicated by individual colors.**

| Excitation (nm) | Blue: 488 | | | | | Red: 633 | | | Violent: 405 | |
|---|---|---|---|---|---|---|---|---|---|---|
| Emission (nm) | 523 | 575 | 613 | 692 | 760 | 650 | 720 | 767 | 455 | 528 |
| Fluorochrome | FITC | PE | ECD | PE-Cy5.5 | PE-Cy7 | APC | AF700 | AA750 | PB | KRO |
| Panel 1 | CD45RA | CCR7 | CD28 | PD1 | CD27 | CD4 | CD8 | CD3 | CD57 | CD45 |
| CC of panel 1 | CD4 | CD4 | CD28 | PD1 | CD27 | CD4 | CD8 | CD3 | CD4 | CD8 |
| Panel 2 | TCRγδ | TCRαβ | HLA-DR | – | TCR Vδ1 | CD4 | CD8 | CD3 | TCR Vδ2 | CD45 |
| CC of panel 2 | CD4 | CD4 | HLA-DR | – | TCR Vδ1 | CD4 | CD8 | CD3 | CD4 | CD8 |

Note:
FITC, Fluorescein isothiocyanate, PE, Phycoerythrin; ECD, Phycoerythrin-Texas Red-X; PE-Cy, Phycoerythrin-Cyanine; APC, Allophycocyanin; AF700, Alexa Flour 700; AA750, Allophycocyanin Alexa Flour 750; PB, Pacific Blue; KRO, Krome Orange; CC, compensation control.

on their postoperative period. In total, 47 patients were 1 year post-transplantation (1-year group), and 41 patients were 5 years post-transplantation (5-year group). All of the patients had experience with dialysis prior to transplantation. Additionally, 41 healthy volunteers were recruited as a control group (healthy group) from our physical examination clinic. All participants provided written informed consent. The study protocol was reviewed and approved by the institutional review board (Ethics Committee) of the 3rd Xiangya Hospital, Central South University (No. 2018-S347).

One mL of peripheral blood was obtained in a vacuum tube (BD, Heidelberg, Germany) containing ethylenediaminetetraacetic acid for anticoagulation. All samples were tested immediately or stored at room temperature for no more than 1 hour after collection.

## Leukocyte staining

Flow cytometric fluorescent anti-human monoclonal cell surface antibody (dry powder) tubes (DuraClone IM) were purchased from Beckman Coulter (Bangalore, India). The details of every fluorochrome-conjugated antibody, the schemes of every fluorochrome channel and the compensation controls (each of a single color) are presented in Table 2. Briefly, 100 μL of anticoagulant blood was stained with fluorescent antibodies for 15 minutes in the dark (room temperature). The erythrocytes were removed by adding two mL of lyse-fix solution consisting of Versa Lyse™ and IOTest^VR Fixative Solution (MBL Life Science, Nagoya, Japan) and incubated for 15 min in the dark (room temperature). Then, the cells were rinsed twice and resuspended in staining buffer (phosphate-buffered saline containing 2% fetal bovine serum) prior to acquisition. All samples were analyzed with a 13-Color CytoFlex Flow Cytometer (Beckman Coulter, Brea, CA, USA) after daily calibration with Flow-Set Pro Beads (Beckman Coulter, Brea, CA, USA).

## Data analysis

The collected flow cytometric information was investigated using the Kaluza Software version 1.2 (Beckman Coulter, Brea, CA, USA) by a single operator according to the ONE-Study protocol (Streitz et al., 2013). For setting up compensation using the AutoSetup Scheduler, refer to the Application Note "Compensation Setup for High Content DuraClone reagents," which is downloadable from the Beckman Coulter

Table 3 Baseline data in different groups (Mean ± SD).

| | 1-year | 5-year | Healthy | p-value |
|---|---|---|---|---|
| Age, years | 35.81 ± 8.69 | 44.53 ± 10.02 | 38.11 ± 9.55 | <0.0001 |
| Gender (Male/Female) | 43/9 | 33/18 | 22/22 | 0.003 |
| WBC ($\times 10^9$/L) | 7.81 ± 2.04 | 7.37 ± 1.20 | 6.71 ± 1.95 | 0.0111 |
| Lymphocytes ($\times 10^9$/L) | 2.02 ± 0.69 | 2.15 ± 0.57 | 1.26 ± 0.43 | <0.0001 |
| Serum creatinine (umol/L) | 117.4 ± 26.76 | 101.88 ± 25.94 | 71.13 ± 17.53 | 0.0015 |
| FK506 mean concentration (ng/mL) | 7.16 ± 1.74 | 5.87 ± 1.54 | – | 0.0002 |

website (https://media.beckman.com/-/media/pdf-assets/application-notes/flow-cytometry-reagents-duraclone-appnote-compensation-setup.pdf). The adhesive doublets were removed by two forward scatter parameters (width vs height). We used CD45 and side scatter to gate leukocytes. The parameters used to calculate the size and frequency of the subsets were exported from the Kaluza software into Excel (Microsoft, Redmond, WA, USA).

## Statistical analysis

The mean ± standard deviation (SD) was used to describe the analyzed data. At baseline, the proportion gender ratio was compared with the Chi-square test of independence, and the mean age and serum creatinine levels were compared by one-way ANOVA. The FK506 concentration was compared by the Mann–Whitney U-test. Differences in T lymphocyte subset percentages among the groups were compared using the Mann–Whitney U-test, because not all of the parameters were distributed normally. GraphPad Prism 7.0 (GraphPad Software Inc., La Jolla, CA, USA) was used to perform the statistical analyses. Values of $p < 0.05$ were considered statistically significant. * indicates *$p < 0.05$, **$p < 0.01$, ***$p < 0.001$, and ****$p < 0.0001$.

## RESULTS

### Demographic and baseline characteristics

The demographic and baseline characteristics of each group are presented in Table 3. The patients in the 5-year group exhibited an increased mean age (44.53 ± 10.02) compared with that of the 1-year group. More male patients were noted in the 1-year and 5-year groups. All participants in our study had peripheral white blood cell counts (95% CI [7.035–7.625]) and lymphocytes counts (95% CI [1.724–1.950]) within the normal ranges. All participants in our study exhibited normal serum creatinine ranges (95% CI [93.20–103.1]). All patients post-kidney transplantation took FK506+MMF+ prednisolone (Pred). The cytomegalovirus (CMV) status was not assessed in this study, because almost all of the enrolled allograft recipients were serologically CMV-positive, and only four of the patients were negative.

### T cell subset gating strategies

Based on the details of each T cell subpopulation shown in Fig. 1, we used the forward scatter area and CD45 to gate peripheral lymphocytes. CD3+ cells were defined as total T cells and divided into five populations using the CD4, CD8, HLA-DR, and TCR cell

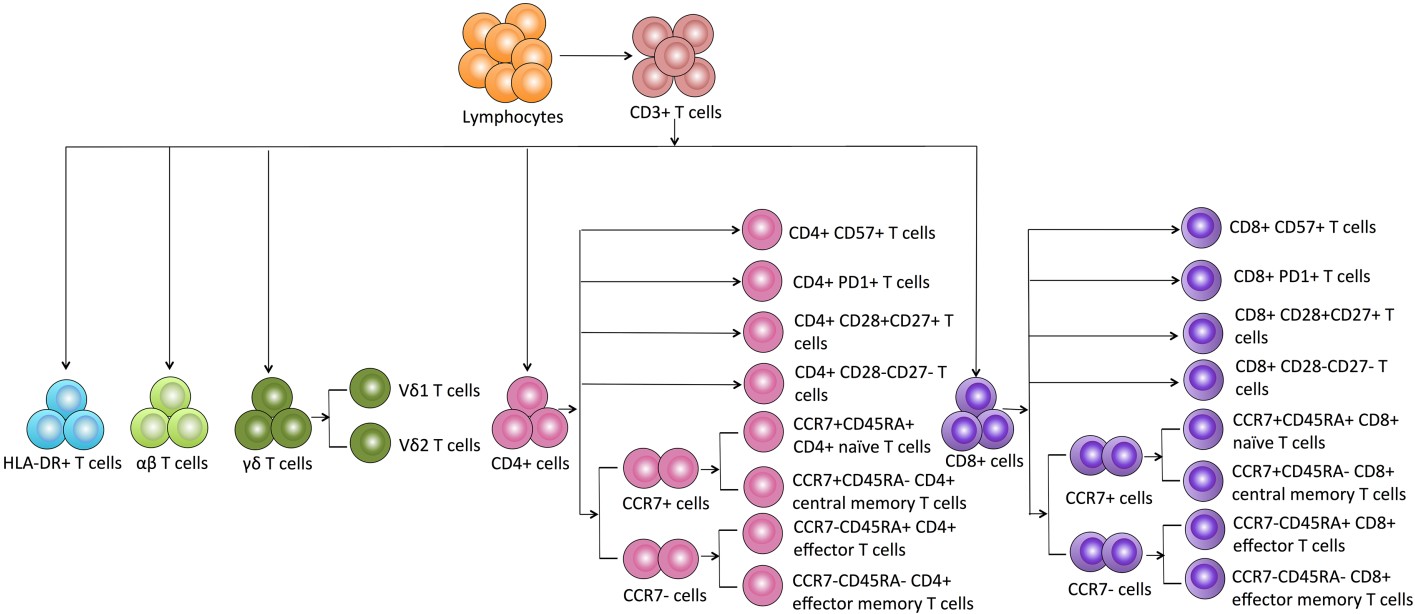

**Figure 1** Schematic overview and flow cytometric gating strategies of T cell subsets in peripheral blood.

markers as follows: (1) CD4+ T cells, (2) CD8+ T cells, (3) HLA-DR+ T cells, (4) αβ T cells, and (5) γδ T cells. The γδ T cells were categorized into Vδ1 and Vδ2 subsets. Additionally, the CD8+ and CD4+ T cells were divided into eight subgroups as follows: CD57+ subset, PD1+ subset, costimulatory molecule (CD28+CD27+ and CD28-CD27-) subsets, and CM (CD45RA-CCR7+), EM (CD45RA-CCR7-), naïve (CD45RA+CCR7+) and effector (CD45RA+CCR7-) cell subsets.

## Distribution of the general T cell subsets

Generally, CD3+ T cells and the two main T cell subsets (CD8+ and CD4+) are tested. We also used HLA-DR to assess the T cell activation status. We observed a significant reduction in the proportion of CD3+ T cells in the 1-year and 5-year kidney transplant patients compared with that of the healthy volunteers. However, no significant difference was noted between the 1-year and 5-year patients. No significant differences in the CD4+, CD8+, and HLA-DR+ T cells were noted among the diverse groups (Fig. 2). Altogether, because only limited differences in general T cell subsets were found between the healthy volunteers and allograft recipients, we needed to identify more detailed T cell subpopulations. All of the means ± SDs and *p*-values are displayed in Table 4.

## Distribution of the TCR T cell subsets

To obtain more details for more specific T cell subsets, we applied TCR markers to define some subsets. We identified a significant decline in the proportion of γδ T cells but an increased proportion of αβ T cells in the renal allograft recipients compared with those of the healthy volunteers. Similar results were observed for the Vδ2 and Vδ1 subsets of the

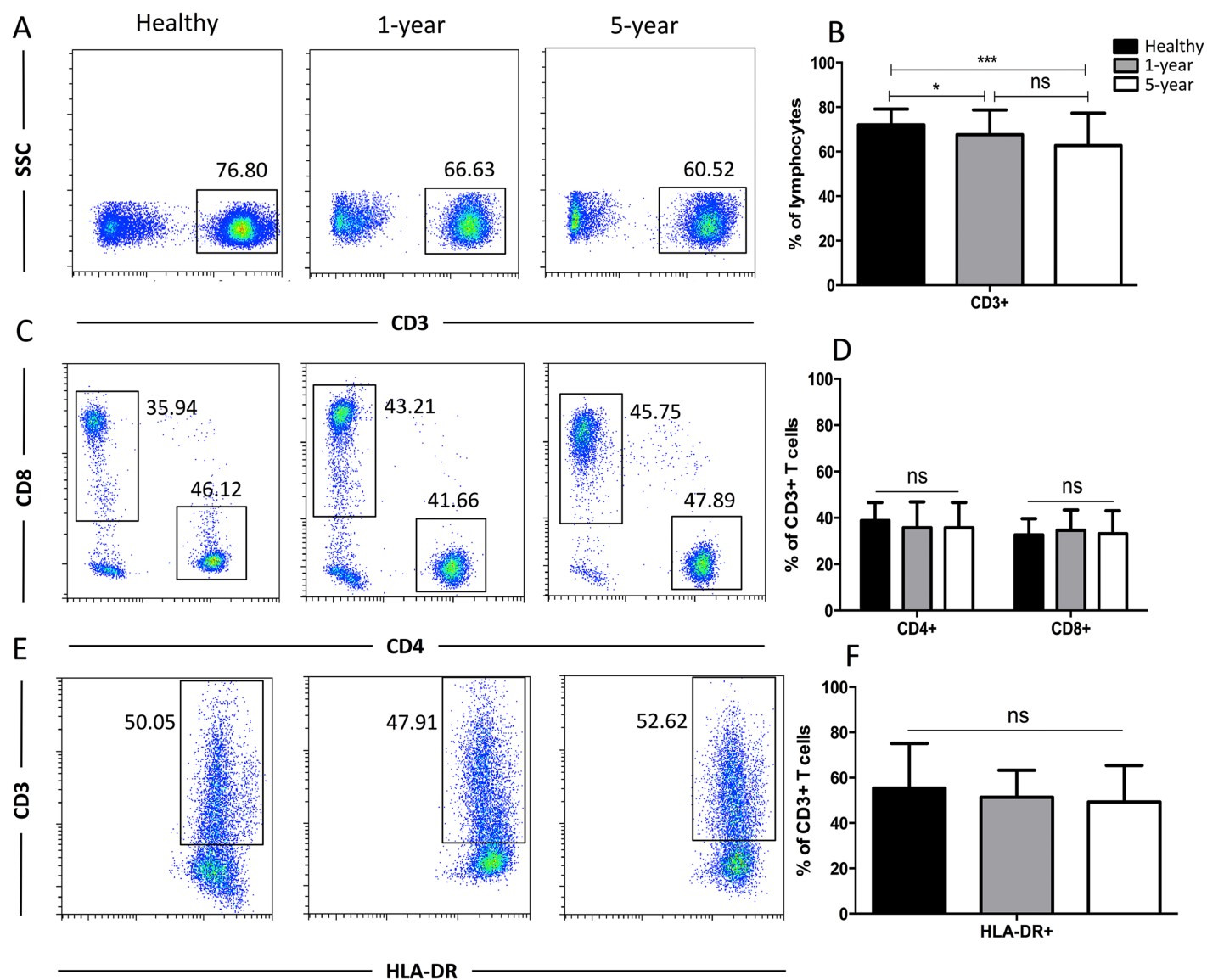

**Figure 2 Parental proportions of total, CD4+, CD8+, HLA-DR+ T cells among different groups.** Healthy individuals showed a higher percentage of total (CD3+) T cells than both 1-year ($p < 0.05$) and 5-year ($p < 0.001$) renal allograft recipients (A) and (B). The differences of CD4+, CD8+, HLA-DR+ T cells were not significant ($p > 0.05$) (C–F). Data are expressed as mean number of each group (mean ± SD). $^{*}p < 0.05$, $^{***}p < 0.001$.

γδ T cells. No significant differences were found in any of the TCR subsets described above between the 1-year and 5-year renal allograft recipients (Fig. 3). Taken together, obvious differences were noted among the TCR T cell subsets between the healthy individuals and kidney transplant recipients. All of the means ± SDs and *p*-values are displayed in Table 4.

## Distribution of the CD57+ and PD1+ T cell subsets

CD57 and PD1 are typical cell surface markers for T cell immune senescence and regulation and thus are also considered good cell surface markers for immunosuppression

**Table 4 The mean, SD and *p*-value of T subsets among healthy volunteers, 1-year and 5-year renal allograft recipients.**

| | Mean ± SD | | | *p*-value | | |
|---|---|---|---|---|---|---|
| | Healthy (H) | 1-year (1y) | 5-year (5y) | H vs 1y | H vs 5y | 1y vs 5y |
| CD3+ | 72.08 ± 7.07 | 67.65 ± 11.07 | 62.88 ± 14.62 | 0.0302 | 0.0005 | 0.0796 |
| CD4+ | 38.77 ± 7.89 | 35.73 ± 11.15 | 35.03 ± 10.91 | 0.1484 | 0.0791 | 0.7687 |
| CD8+ | 32.63 ± 6.97 | 34.64 ± 8.72 | 33.15 ± 9.88 | 0.2410 | 0.7829 | 0.4559 |
| CD3+HLA-DR+ | 55.38 ± 19.77 | 51.37 ± 11.91 | 49.28 ± 16.09 | 0.2457 | 0.1291 | 0.4858 |
| αβ | 85.22 ± 11.41 | 90.46 ± 6.09 | 91.25 ± 6.16 | 0.0076 | 0.0038 | 0.5474 |
| γδ | 14.20 ± 11.39 | 8.99 ± 5.82 | 8.04 ± 6.04 | 0.0072 | 0.003 | 0.4577 |
| Vδ1 γδ | 17.28 ± 17.87 | 35.47 ± 16.22 | 35.81 ± 23.13 | <0.0001 | 0.0001 | 0.9354 |
| Vδ2 γδ | 73.96 ± 22.10 | 43.96 ± 22.13 | 46.52 ± 28.61 | <0.0001 | <0.0001 | 0.6374 |
| CD4+ CD57+ | 3.77 ± 2.76 | 8.34 ± 8.44 | 4.31 ± 3.63 | 0.0014 | 0.4540 | 0.0057 |
| CD4+ PD1+ | 32.36 ± 9.09 | 38.47 ± 12.87 | 38.60 ± 15.89 | 0.0105 | 0.0271 | 0.9652 |
| CD8+ CD57+ | 36.71 ± 13.36 | 41.20 ± 13.44 | 42.59 ± 14.90 | 0.1212 | 0.0636 | 0.6453 |
| CD8+ PD1+ | 25.60 ± 8.81 | 34.17 ± 13.50 | 32.34 ± 13.25 | 0.0008 | 0.0082 | 0.9240 |
| CD4+ CD28+CD27+ | 86.59 ± 4.63 | 74.31 ± 11.74 | 82.92 ± 7.44 | <0.0001 | 0.0088 | 0.0001 |
| CD4+ CD28-CD27- | 5.22 ± 2.73 | 15.01 ± 9.85 | 7.15 ± 7.11 | <0.0001 | 0.1073 | <0.0001 |
| CD8+ CD28+CD27+ | 50.02 ± 13.34 | 36.21 ± 12.61 | 34.53 ± 14.48 | <0.0001 | <0.0001 | 0.5620 |
| CD8+ CD28-CD27- | 34.74 ± 12.13 | 47.64 ± 12.60 | 52.32 ± 15.63 | <0.0001 | <0.0001 | 0.1238 |
| CD4+ CM T | 45.53 ± 8.38 | 40.13 ± 12.96 | 56.23 ± 10.87 | 0.0250 | <0.0001 | <0.0001 |
| CD4+ EM T | 15.02 ± 4.82 | 25.74 ± 15.14 | 22.76 ± 10.40 | <0.0001 | <0.0001 | 0.2933 |
| CD8+ CM T | 7.56 ± 3.66 | 4.93 ± 4.59 | 5.71 ± 4.29 | 0.0043 | 0.0394 | 0.4157 |
| CD8+ EM T | 31.37 ± 9.86 | 28.17 ± 12.71 | 26.07 ± 13.08 | 0.1945 | 0.0415 | 0.4489 |

and tolerance, respectively. In the CD4+ subsets, the percentage of CD57+ T cells was highest in the 1-year renal allograft recipients compared with those of the healthy individuals and 5-year recipients. No significant difference was found between the healthy volunteers and 5-year renal allograft patients. Additionally, no significant differences were noted in the CD8+ CD57+ T cells among the groups. The percentages of PD1+T cells in both the CD4+ and CD8+ populations were significantly increased in the renal allograft recipients compared with those of the healthy volunteers. Nevertheless, no significant difference was found between the 1-year and 5-year renal allograft recipients (Fig. 4). All of the means ± SDs and *p*-values are displayed in Table 4.

## Distribution of the costimulatory molecule T cell subsets

In the costimulatory molecule (CD27 and CD28) subsets, only the CD27 and CD28 double-positive and double-negative subsets exhibited significant differences. The percentages of CD27+CD28+ T cells in both the CD4+ and CD8+ populations were obviously decreased in the renal allograft recipients compared with those of the healthy volunteers. The CD4+ CD27+CD28+ T cells were reduced in the 1-year compared with the 5-year recipients. In contrast, the percentages of CD27 and CD28 double-negative T cells in both the CD4+ and CD8+ populations were significantly increased in the renal allograft recipients compared with those of the healthy volunteers. CD27 and CD28

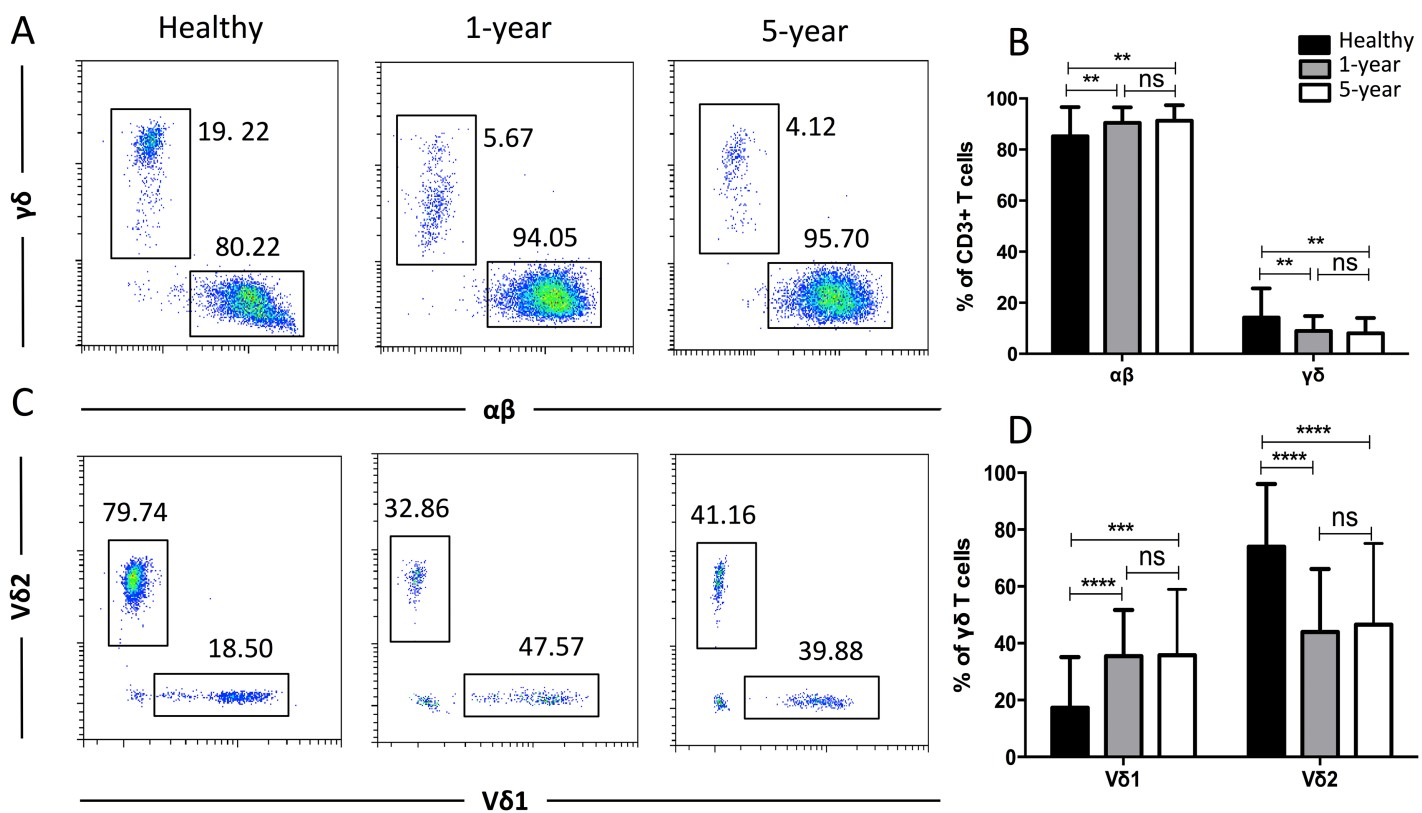

**Figure 3 Parental proportions of αβ, γδ, and Vδ1 and Vδ2 γδ T cells among different groups.** Healthy individuals showed a lower percentage of αβ T cells, but a higher percentage of γδ T cells than both 1-year ($p < 0.01$) and 5-year ($p < 0.01$) renal allograft recipients (A) and (B). Healthy individuals also showed a lower percentage of Vδ1 but a higher percentage of Vδ2 γδ T cells than both 1-year ($p < 0.0001$) and 5-year ($p < 0.0001$) renal allograft recipients (C) and (D). The differences between 1-year and 5-year recipients from each TCR subsets above were not significant ($p > 0.05$) (A–D). Data are expressed as mean number of each group (mean ± SD). **$p < 0.01$, ****$p < 0.0001$.

double-negative CD4+ T cells were increased in the 1-year over the 5-year recipients. No obvious differences in both the CD27 and CD28 double-negative and -positive T cells in the CD8+ subsets were noted between the 1-year and 5-year renal allograft recipients (Fig. 5). All of the means ± SDs and *p*-values are displayed in Table 4.

## Distribution of the memory T cell subsets

We observed a significantly increased percentage of CD4+ EM T cells but a decreased percentage of CD8+ CM T cells in the renal allograft recipients compared with those of the healthy volunteers. However, the differences in the above memory T cell subsets were not significant between the renal allograft recipients and healthy volunteers (Fig. 6). All of the means ± SDs and *p*-values are displayed in Table 4.

## DISCUSSION

In our study, we used multicolor flow cytometry to analyze the detailed T cell subpopulation immunophenotyping of peripheral blood specimens from 88 renal transplant patients 1 year and 5 years after transplantation and compared these results

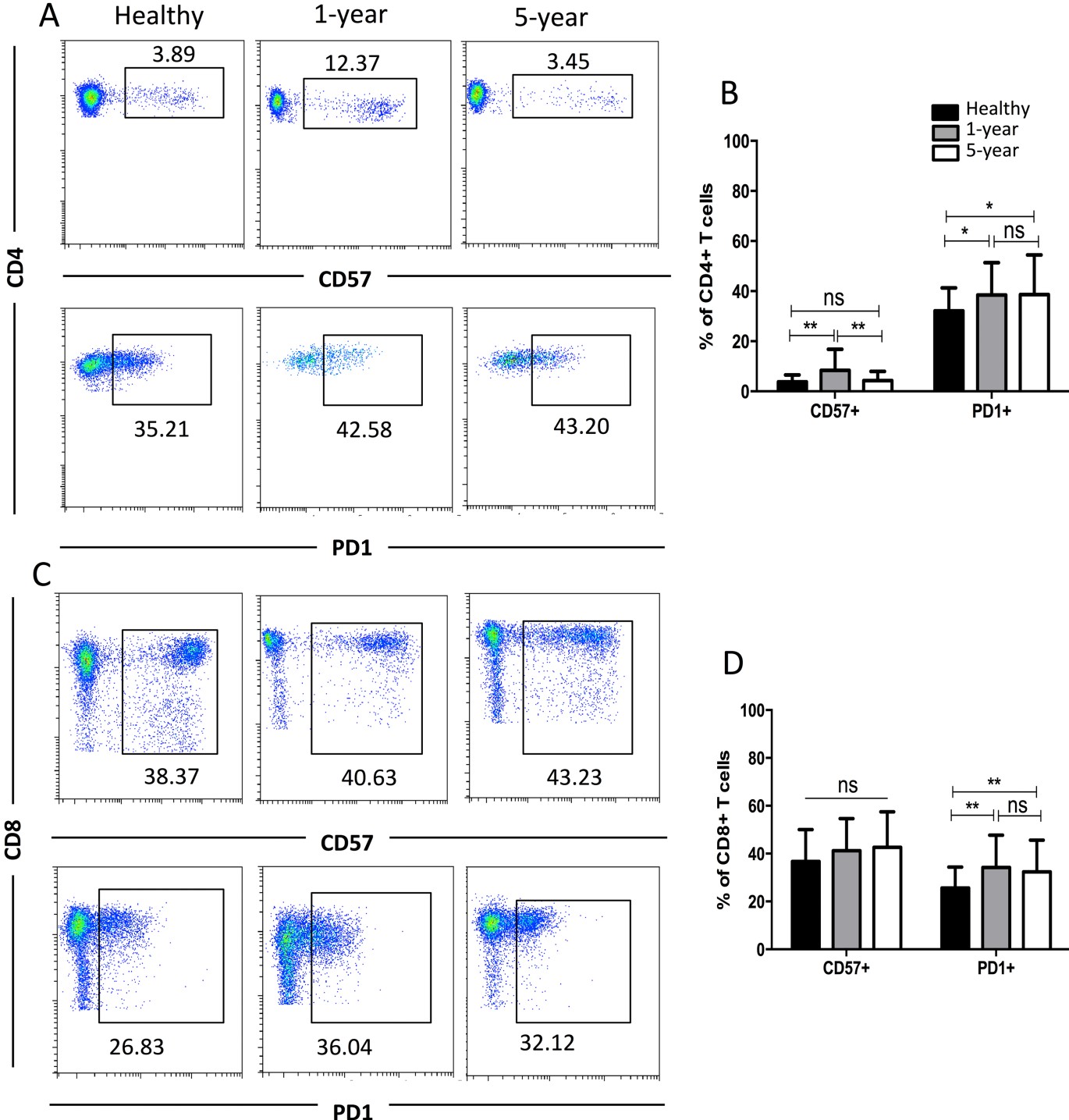

**Figure 4 Parental proportions of CD57+ and PD1+ T cells among different groups.** In CD4+ T cells, the percentage of CD57+T cells was the highest in 1-year renal allograft recipients compared with healthy individuals ($p < 0.01$) and 5-year recipients ($p < 0.01$). No significant difference was addressed between healthy individuals and 5-year renal allograft patients ($p > 0.05$). The percentage of PD1+T cells was significantly increased in renal allograft recipients than healthy individuals ($p < 0.05$). No significant difference was addressed between 1-year and 5-year renal allograft patients ($p > 0.05$) (A) and (B). In CD8+ T cells, no significant difference in CD57+ T cells was noted among all the three groups ($p > 0.05$). The percentage of PD1+T cells populations was significantly increased in renal allograft recipients than healthy individuals ($p < 0.05$). No significant difference was addressed between 1-year and 5-year renal allograft patients ($p > 0.05$) (C) and (D). Data are expressed as mean number of each group (mean ± SD). $^{*}p < 0.05$, $^{**}p < 0.01$.

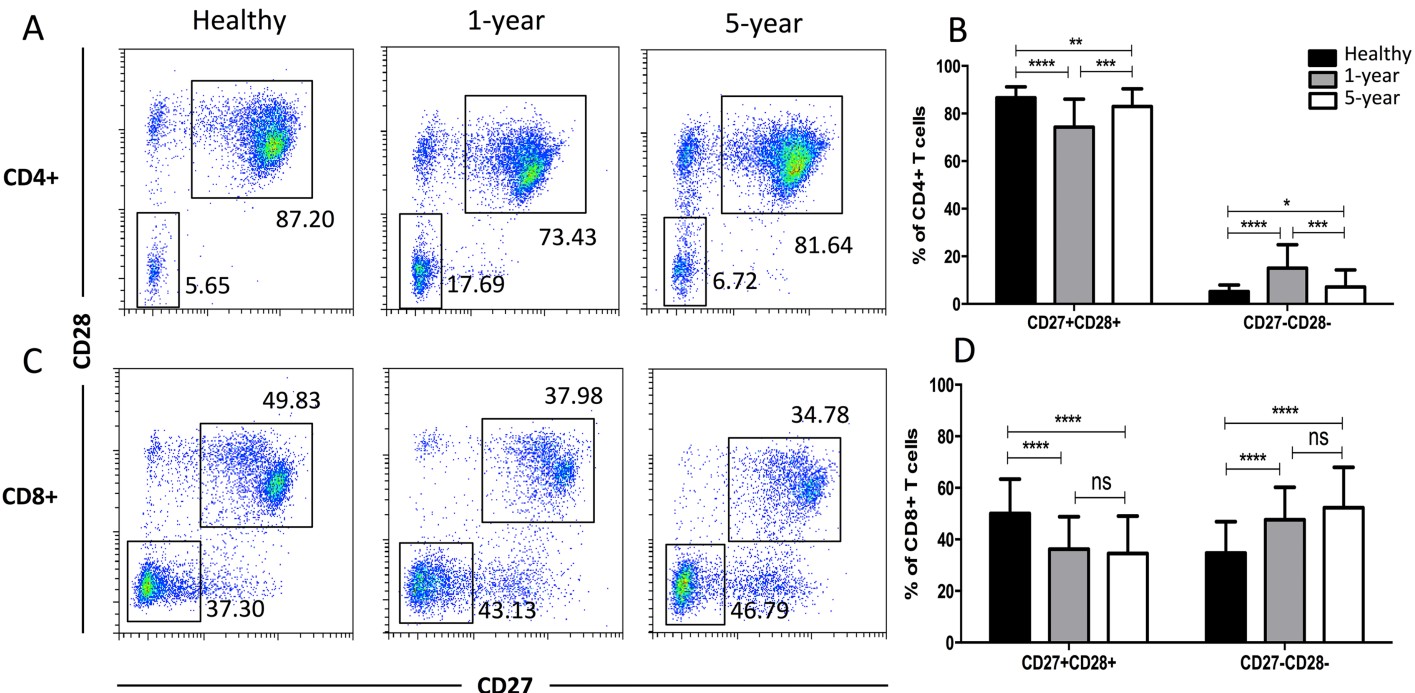

**Figure 5 Parental proportions of costimulatory molecular (CD27 and CD28) T cells among different groups.** In CD4+ subsets, healthy individuals showed a higher percentage of CD27+CD28+ T cells than both 1-year ($p < 0.0001$) and 5-year ($p < 0.01$) renal allograft recipients; 1-year recipients had a lower percentage than 5-year group ($p < 0.001$). Healthy individuals showed a lower percentage of CD27-CD28- T cells than both 1-year ($p < 0.0001$) and 5-year ($p < 0.05$) renal allograft recipients; 1-year recipients had a higher percentage than 5-year group ($p < 0.001$) (A) and (B). In CD8+ subsets, healthy individuals showed a higher percentage of CD27+CD28+ T cells than both 1-year ($p < 0.0001$) and 5-year ($p < 0.0001$) renal allograft recipients. Healthy individuals showed a lower percentage of CD27-CD28- T cells than both 1-year ($p < 0.0001$) and 5-year ($p < 0.0001$) renal allograft recipients (C) and (D). The differences of both CD4+ and CD8+ CD27-CD28- T cells were not significant ($p > 0.05$) (A–D). Data are expressed as mean number of each group (mean ± SD). $^{*}p < 0.05$, $^{**}p < 0.01$, $^{***}p < 0.001$, and $^{****}p < 0.0001$.

with those of healthy volunteers. The distribution of T cell subpopulations in immune-stable allograft recipients was identified.

The fraction of total T (CD3+) cells declined in the renal allograft recipients. The percentage of CD8+ T cells exhibited an increasing trend in the renal allograft recipients compared with that of the healthy volunteers; however, the difference was not statistically significant. HLA-DR expression, which might be activation-dependent, was also assessed (*Arneth, 2018*). However, in our study, no significant differences in HLA-DR+ T cell subsets were noted. This finding may be attributed to triggering of a few T cells in the immune-stable patients and healthy volunteers.

γδ T cells are a small subsection (3–5%) of T cells in human peripheral blood. According to their TCR variable (V) gene fragment, there are two main subgroups (Vδ1 and Vδ2). Vδ2 T cells are the major γδ T cell population in circulation, accounting for 50–95% of the γδ T cells in peripheral blood mononuclear cells (*Peters, Kabelitz & Wesch, 2018*). Although some evidence indicates that γδ T cells can apply immunosuppressive features, the majority of γδ T cell activities are pro-inflammatory immune responses (*Tyler et al., 2015; Xu et al., 2018*). Furthermore, Vδ2 T cells can give rise to Th1-, Th2- (*Wesch, Glatzel & Kabelitz, 2001*), Th9- (*Peters et al.,*

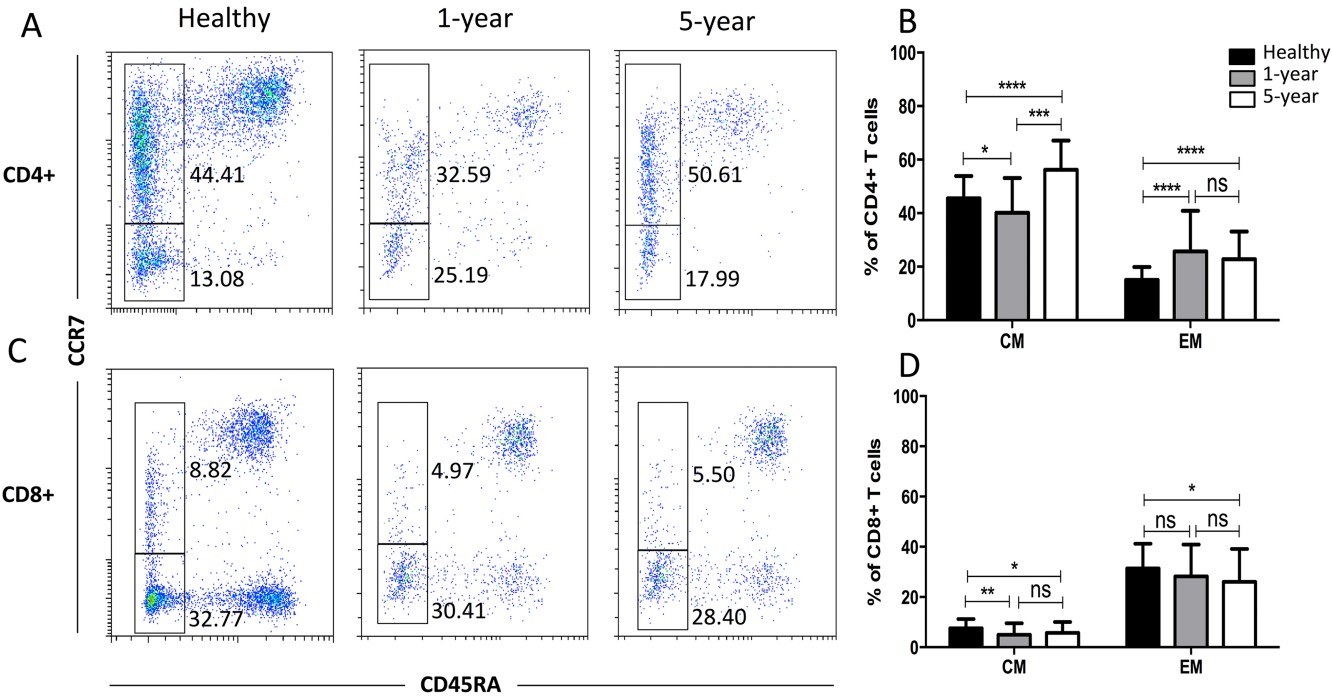

**Figure 6  Parental proportions of memory T cells among different groups.** In CD4+ T cells, 5-year recipients showed a higher percentage of CM T cells than both 1-year ($p < 0.001$) and healthy individuals ($p < 0.0001$); 1-year recipients had a lower percentage than healthy individuals ($p < 0.05$). Healthy individuals showed a higher percentage of EM T cells than both 1-year ($p < 0.0001$) and 5-year ($p < 0.0001$) renal allograft recipients, and no significant difference was addressed between 1-year and 5-year renal allograft patients ($p > 0.05$) (A) and (B). In CD8+ T cells, healthy individuals showed a higher percentage of CM T cells than both 1-year ($p < 0.01$) and 5-year ($p < 0.05$) renal allograft recipients, and no significant difference was addressed between 1-year and 5-year renal allograft patients ($p > 0.05$). Healthy individuals showed a higher percentage of EM T cells than 5-year renal allograft recipients ($p < 0.05$). The differences of CD8+ EM T cells were not significant both between healthy individuals and 1-year group and between 1-year and 5-year groups ($p > 0.05$) (C) and (D). Data are expressed as mean number of each group (mean ± SD). $^*p < 0.05$, $^{**}p < 0.01$, $^{***}p < 0.001$, and $^{****}p < 0.0001$.

*2016*), Th17- (*Caccamo et al., 2011*), Tfh- (*Bansal et al., 2012*), and APC-like phenotypes (*Brandes, Willimann & Moser, 2005*). These findings potentially explain why significantly decreased proportions of γδ and Vδ2 T cells were noted in the immune-stable renal allograft recipients. Additionally, we presented the percentages of Vδ2- and Vδ1-positive cells as a percentage of the CD3+ cells and not simply as a percentage of the γδ T cells. We found that the percentage of CD3+ Vδ2 cells in the controls was significantly higher than that in the allograft recipients. However, no significant difference in CD3+ Vδ1 cells was noted among the three groups (Fig. S1).

CD57 was first reported as a marker of natural killer cells (*Kared et al., 2016*). CD57 is present in CD8+ and CD4+ T cells at the late stages of differentiation and usually is applied for identification of terminally differentiated "senescent" cells with a lower proliferative ability and altered characteristics (*Brenchley et al., 2003*). In this paper, the proportion of CD4+CD57+ T cells was increased in the renal allograft recipients, indicating that more "senescent and exhausted" T cells were present in the immune-stable allograft recipients. Moreover, patients in the 1-year group exhibited an increased proportion of CD4+CD57+ T cells compared with that of the 5-year group, demonstrating

more "senescent" T cells in the short-term patients post-kidney transplantation. CD279 (PD1) makes a significant contribution to the balance of T cell immunity and immune tolerance and binds its ligand (PDL1) to induce T cell apoptosis (*Mahoney, Freeman & McDermott, 2015*; *Zhang et al., 2016*). In this study, the proportions of both CD8+ and CD4+ PD1+ T cells were increased in the renal allograft recipients, suggesting that more reactive T cells were potentially undergoing apoptosis and reversible exhaustion in the immune-stable allograft recipients. Additionally, the terminal effector stages of T cell differentiation are indicated by upregulation of CD57 (effector phenotype) and PD1 (coinhibitory molecule, exhausted phenotype) expression (*Booiman et al., 2017*), which basically is consistent with the above results.

As secondary signals, the T cell costimulatory molecules CD27 and CD28 play very pivotal roles in T cell full activation (*Tanaskovic et al., 2017*). Many immune anergy and suppression therapeutic strategies have focused on these molecules. In our study, CD27+CD28+ (costimulatory molecule double-positive) T cells in the CD4+ and CD8+ groups were reduced in the renal allograft recipients, whereas CD27-CD28- (double-negative) T cells were increased. Moreover, CD4+CD27+CD28+ T cells were decreased but CD4+CD27-CD28- T cells were increased in the 1-year group patients compared with those of the 5-year group patients, indicating that T cell costimulatory signals were reduced in the immune-stable allograft patients in the short term. This result might be consistent with a recent study showing that immunological aging-related expansion of highly differentiated CD28- T cells was associated with higher immunosuppression (*Dedeoglu et al., 2016*).

The traditional T cell subpopulations of naive, CM (CCR7+CD45RA-), EM (CCR7- CD45RA-) and effector T cells were first defined based on CCR7 and CD45RA expression (*Maecker, McCoy & Nussenblatt, 2012*). Compared with those of naive T cells, memory T cells require lower activation conditions and can rapidly induce alloimmune responses through synthesis of a variety of inflammatory cytokines and cytolytic effectors (*Adams et al., 2003*). CD4+ EM T cells are linked to the occurrence of acute cellular and antibody-mediated rejection (*Danger, Sawitzki & Brouard, 2016*). Despite taking immunosuppressive drugs, renal allograft recipients still show a danger of acute cellular and antibody-mediated rejection. Therefore, in our study, an increased fraction of CD4+ EM T cells was noted in the renal allograft recipients compared with that of the healthy volunteers. Additionally, CD8+ CM T cells were decreased in the renal allograft recipients in our study. Given that investigations into the role of CCR7 in transplant processes have yielded conflicting results (*Ziegler et al., 2006*), explaining the exact mechanism of this phenomenon is difficult. Dedeoglu and his colleagues also investigated these memory T subsets in both the peripheral blood and lymph nodes. They found that the median frequencies of CD4+ EM and CD4+CD28null T cells were significantly higher within patients with allograft rejection, but no other significant differences were observed for the other CD4+ and CD8+ T cell subsets (*Dedeoglu et al., 2017*). More functional studies should focus on this subset in transplant patients. Additionally, highly differentiated memory T cells are characterized by loss of the costimulatory molecule CD28, making them less dependent on costimulation to become
activated (*Weng, Akbar & Goronzy, 2009*). Therefore, we also examined terminally differentiated T cells. We analyzed the percentages of CM and EM cells in the CD28-positive and -negative cell populations. We found that the frequency of CD4+CD28+EM cells in the renal recipients was significantly higher and the frequency of CD8+CD28-CM cells was significantly lower than those in the healthy volunteers (Fig. S2). These differences were consistent with the CD4+ EM and CD8+ CM cell results described above.

Interestingly, we found that the CD8+ T cells were expressed at low levels in our study, which might have been the result of recent TCR stimulation, as indicated by the downregulation of CD8. Therefore, we analyzed the CD8low subsets to observe whether they varied between the healthy volunteer and patient groups. We found that the frequencies of CD57+ and CD27-CD28- subsets from the CD8low T cells were significantly higher in the patient groups than in the healthy volunteers. However, the frequencies of the PD1+, CD27+CD28+, CM, and EM subsets from the CD8low T cells were significantly lower in the patient groups than in the healthy volunteers (Fig. S3). The results were not completely consistent with those obtained for the total CD8+ T subsets.

Renal failure and dialysis are known to change a patient's immune profile, which leads to T cell dysfunction (*Betjes, 2013*). Although renal function in the enrolled subjects was corrected with transplantation, whether the transplant also reverted the immune dysregulation was unclear. Therefore, elucidating the impact of previous ESRD on the differences is important and necessary. We enrolled uremia patients aged from 18 to 65 years who had undergone dialysis in our department. None of these patients had received immunosuppressive treatment. Some useful information and results were found compared with those of the healthy volunteers and renal allograft recipients. (1) The frequency of CD8+ T cells was significantly lower than those in the other groups, which was consistent with the description of Costa's and Cheng's studies (*Cheng, Chen & Li, 1991*; *Costa et al., 2008*). (2) The frequency of the CD3+ HLA-DR+ T cell population (activated T cells) in the uremia patients was significantly lower than that in the healthy volunteers, and the frequencies of both the 1-year and 5-year recipients were also significantly lower than that of the uremia patients. (3) In the TCR αβ and γδ subgroups, the frequency of αβ T cells in the uremia patients was significantly higher and the frequency of γδ T cells was significantly lower than those in the other groups. In a recent study, significant inhibition of the γδ T cell population was demonstrated in patients with ESRD (*Juno et al., 2017*). (4) The frequencies of both CD4+ and CD8+ CM cells were significantly higher but the frequencies of both CD4+ and CD8+ EM cells were significantly lower than those of the other groups (Fig. S4). A previous study showed that the percentages of CM and EM T cells were significantly higher in the ESRD group than in the healthy group (*Chung et al., 2012*). However, in our study, both CD4+ and CD8+ EM T cells showed a lower frequency in the ESRD group. Although this finding was not consistent with Chung's observation, the ratio of EM/CM T cells was decreased in the ESRD group, which was identical to the conclusion of Segundo's study (*Segundo et al., 2010*). Taken together, an impact of previous renal disease and dialysis indeed existed in some T cell subsets. However, we could also speculate, albeit not

strongly, that kidney transplantation might revert not only renal function but also T cell immune dysregulation.

As stated in the beginning, most renal allograft recipients reach a stable immune state (neither rejection nor infection) after transplantation. The importance of this study is that we provide an overview of the stable immune state of renal allograft recipients. We evaluated the immune state of renal allograft recipients based on normal immunity in the past, but renal allograft recipients also have their own stable immune state. We wanted to identify differences between this stable immune state and a normal immune state. In the future, we may perform a large sample-size study to provide the basic background and criteria for renal allograft recipient immunity and conduct further studies to evaluate immune cell changes in response to infection, rejection and other states according to stable immunity instead of normal immunity. For this purpose, we should elucidate the differences, and thus a large sample size study may be warranted to define the criteria.

## CONCLUSION

We comprehensively evaluated the immune state of stable renal allograft recipients in detail and found that the distribution of most T cell subpopulations in immune-stable renal allograft recipients significantly differed from those of healthy volunteers, including γδ, Vδ2, PD1, CD27+CD28+, CD27-CD28-, CD4+ EM, and CD8+ CM cells. The proportion of some of these T cell subsets in the renal allograft recipients also differed in the short-term compared with those in the long-term. However, more detailed information should be included in subsequent studies, such as the absolute numbers of each T cell subset and the proportion of regulatory T cells.

### Funding

This study was supported by grants of the National Natural Science Foundation of China (81700658 and 81771722), the Natural Science Foundation of Hunan Province (2016JJ4105), the New Xiangya Talent Project of the Third Xiangya Hospital of Central South University (JY201629), and the University Student Innovation Program of Central South University (ZY20180983 and ZY20180988). The funders had no role in study design, data collection and analysis, decision to publish, or preparation of the manuscript.

### Grant Disclosures

The following grant information was disclosed by the authors:
National Natural Science Foundation of China: 81700658 and 81771722.
Natural Science Foundation of Hunan Province: 2016JJ4105.
New Xiangya Talent Project of the Third Xiangya Hospital of Central South University: JY201629.
University Student Innovation Program of Central South University: ZY20180983 and ZY20180988.

## Competing Interests

The authors declare that they have no competing interests.

## Author Contributions

- Quan Zhuang conceived and designed the experiments, performed the experiments, analyzed the data, prepared figures and/or tables, authored or reviewed drafts of the paper.
- Bo Peng performed the experiments, analyzed the data, contributed reagents/materials/analysis tools.
- Wei Wei analyzed the data, contributed reagents/materials/analysis tools.
- Hang Gong performed the experiments, analyzed the data, contributed reagents/materials/analysis tools.
- Meng Yu performed the experiments.
- Min Yang performed the experiments.
- Lian Liu performed the experiments.
- Yingzi Ming conceived and designed the experiments, approved the final draft.

## Human Ethics

The following information was supplied relating to ethical approvals (i.e., approving body and any reference numbers):

The study was approved by the Ethics Committee of the 3rd Xiangya Hospital of Central South University and the methods were carried out in accordance with the approved guidelines of the institution (approval: 2018-S347). Written informed consent was obtained from all participants.

## Data Availability

The raw measurements are provided in the Supplemental Files.

## Supplemental Information

Supplemental information for this article can be found online at http://dx.doi.org/10.7717/peerj.6417#supplemental-information.

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
