# Peer review of "The detailed distribution of T cell subpopulations in immune-stable renal allograft recipients: a single center study"

_PeerJ, doi:10.7717/peerj.6417_

## Round 0.1 · original submission · Major Revisions

Please address the concerns with additional experiments as necessary. It will be key to address the key issue brought up by the reviewer, namely, a critical control group to determine if the differences the authors find is due to previous renal disease or immunosuppression. Also, please highlight how this work is distinguished from other efforts in the literature.

Reviewer 1 ·

Basic reporting

1. All figures need legends.

2. Introduction could be imporved with significantly more detail in the description of the cell populations studied. For instance, the relevance of naïve, effector memory, central memory, and effector to the immune state after transplantation is not described. PD1 is listed as important because it is “an immune regulatory marker,” which does not convey the way in which it is immune regulatory. The statement about CD57 is correct, but lacking in detail. For instance, CD57+ cells tend to have proliferative defects but maintain cytokine production (Immunology. 2011 Sep; 134(1): 17–32. doi: 10.1111/j.1365-2567.2011.03470.x).

3. Clarity could be added by conversion of the lists of mean +/- error bars in the text (i.e. lines 214-218 and lines 248-250) to table format.

4. The figures are relevant to the experimental question and laid out well, but a few features need to be altered to improve clarity. Specifically,
- As noted above, figures should have both titles and legends.
- In Figure 2, on the representative flow plots of CD3 by HLA-DR, either the axes are swapped or the conclusion is misstated.

5. It is believed that English is not the first language for the authors and the paper would benefit from an English editor. A few specific examples are:
- In the text that describes Figure 1, there is a statement that “there were 4 groups both in CD8+ and CD4+ T cell..” This is confusing because following this statement 6 separate groups are described, and in the figure shows 10.
- Table 1- typo “teat” should be test.
- Lines 268-270 bring in discussion of pre-formed allogeneic cells but the following paragraph does not address this.
- Using the phrase “normal people” to describe the controls implies that transplant recipients are not normal. “healthy volunteer” may be a better phrase.

6: there are 87 rows of data in T_sub_stat_data and 46 in TCR_stat data but the methods section notes 103 transplant recipients and 44 healthy controls. Why are there fewer subjects in the supplemental data.

7: Table 1 is too vague– what is meant by generalized infection? When were patients assessed for HIV, HCV, HBsAg. It should be mentioned if patients were on immunosuppression other than tacrolimus.

8: Table 3 – should have p values.

Experimental design

1: The research question is well-defined in the focus on T cell phenotyping after transplantion. However, as presented the experiments are less within the scope of the second portion of the question, about evaluation of immunosuppression. Specifically, the authors demonstrate that the FK506 level is higher in the 1 year post-transplant group than the 5 year post-transplant group, but the immune parameters studied for the most part only show a difference between post-transplant and normal. Thus, the authors have not proven that immunosuppression state is the reason for the difference observed ; correlation but not causation is indicated by the data. Conclusion needs to be toned down.

2: Important details of the analysis are missing from the methods section. Specifically:
- There is no description of compensation controls used in the experiments. The data appear to be compensated, but without a description of the compensation method the quality thereof cannot be evaluated. A notation of cell compensation versus beads would be sufficient.
- The flow cytometer used is not mentioned, so its appropriateness also cannot be evaluated.
- Fluorescence minus one controls are listed as being used to set the gates, but again, in the absence of the data being show these are difficult to evaluate.
- The data in the representative figures of flow cytometry data include instances where there is data adjoining the X axis (Figure 5 top, Figure 6 left), and it is unclear how the gating was completed to factor in the data on the axis. For accurate quantification, the gates must include all events along the axis.

3: The methods section specifies two different statistical tests used, and in the absence of figure legends it is impossible to know which test was used for each figure and thus whether an appropriate test was used. In general, the Mann-Whitney test appears to be the more appropriate test.

4: Because of the missing experimental details in the current manuscript, it would be difficult to replicate the study.

5: Ethics appear to be appropriate given the consent form submitted, but a more detailed statement of the ethics safeguards in the methods section as opposed to only in the supplemental materials would be appropriate.

6: No patients received lymphodepletion so any conclusions need to be limited to this group. Similarly, as this is a single center study of Chinese transplant recipients, conclusions should be limited to this group.

Validity of the findings

1: CMV serostatus is known to affect the T cell populations described ( for example, Science Reports 2016. 6:26892 doi: 10.1038/srep26892). Thus, some of the differences observed (or lack of differences between 1 yr and 5 yr) could be related to CMV status. Table 3 should include a column of information on proportion of patients who are CMV seropositive vs seronegative, and if there is a mix, the data may need to be divided between the two groups to determine whether there are differences between the time points when controlling for serostatus.

2: The conclusion that T cell populations are different in transplant recipients relative to healthy volunteers is well supported, but the conclusions about differences short- and long-term after transplant and about potential markers of immunosuppression are speculative. Additionally, some of the conclusions about specific parameters require additional justification from the literature:
- CD8 T cells are listed in the discussion as “suppressive” T cells. Subsets of CD8 T cells can be suppressive, but on the whole these are cytotoxic cells that attack cells that are infected or appear to be foreign.
- CD57 is listed as being associated with senescence and exhaustion. CD57 is also associated with altered function, but it is unclear whether it actually contributes to senescence (Larbi and Fulop, Cytometry Part A. 85(1)25-35. doi: https://doi.org/10.1002/cyto.a.22351)
- PD1 is listed as being associated with T cell apoptosis, and the authors suggest that therefore the high PD1 in transplant recipient samples suggests apoptosis. However, PD1 is also associated with reversible exhaustion (see above reference), so it’s unclear why the hypothesis here is that it is associated with apoptosis. Also, PD1 throughout is listed as being associated with immune regulation, without any definition of what immune regulation means in this context.

3: Controls of the data appear to be appropriate, though as discussed above for research question, they are appropriate specifically for the question of differences between healthy and transplant populations, and less so for identifying effects of immunosuppression levels. The majority of the conclusions focus on time post transplant, but some (particularly in abstract summary) suggest that the data provide information about FK506 dose specifically, which is not the case.

4: several of the conclusions are highly speculative and should be more clearly identified as such, or more analysis should b completed to demonstrate their validity.

5: The CD8 T cells analyzed include CD8low cells, which may have had recent TCR stimulation, as indicated by the CD8 downregulation. Thus, this T cell population could vary between the healthy volunteer and patient groups, and some of the other parameters could vary within it.

Additional comments

The manuscript “The detailed distribution of T cell subpopulations in immune-stable renal allograft recipients” discusses the use of T cell phenotyping to describe differences between healthy and stable post-transplant immune states, with the goal of assessing immunosuppressive state. This research would be of value to publish in PeerJ, but many significant details are missing that would be required for it to be acceptable for publication. However, all of these details should be available to the researchers without further experiments.

Reviewer 2 ·

Basic reporting

The English grammar usage is very good.

The authors have performed a phenotypic analysis of peripheral blood T cells from normal controls and from kidney transplant patients 1 and 5 years out post-transplant. Most of the data presented has been previously published for most of the markers analyzed in the current data set. The authors should do a better job in citing these previous studies; for example Carla Baan and her collaborators have done quite a bit of such phenotypic analysis of T cells from kidney transplant patients.

The structure of the article is professional with no concerns.

Experimental design

No comment.

Validity of the findings

There are a couple of places where further analysis is warranted.

1) For the data in Figure 3, the authors should present the percentage of Vd2 and Vd1 positive cells as a percentage of the CD3+ cells not just as a percentage of the gd+ cells. If they do this they will see that the Vd2+ cells in the control are approximately 16% of the CD3+ cells whereas for the kidney transplant patients they are less than 2% of the CD3+ cells.

The data in 5 raises questions about gating. The CD4+ cells in the 5 year post-kidney transplant patients is placed far to the left of the other two groups, particularly for the analysis of CD57. There is also a problem with the x-axis labeling on the lower left panel. Why do the authors think that their anti-PD-1 stains the CD8 T cells so distinctly but stains the CD4 T cells as a smear?

Studies from several groups have analyzed the expression of CD28 and other exhaustion markers on non-naive/CD45RO+ cells in transplant and cancer patients. The authors should also perform and present this analysis to determine percentages of CD28+ and CD28- cells in the CM and EM cell populations.

Reviewer 3 ·

Basic reporting

The manuscript is well written with only occasional typos and a few less clear sentences. For example, in the METHOD section, the authors stated "all participants received written informed consent" when it should be " all participants provided or gave written informed consent".

References are appropriate and background are sufficient.

The manuscript is professionally structured and the data presentation is very clear. The authors listed tables and figures as subtitles throughout the manuscript which is rather unconventional. They should be referenced in the text without the captions. Analyzed flow cytometry data were provided as an excel workbook, although the true raw data should the the fcs files themselves, which may not be what PeerJ is looking for.

The manuscript is self-contained, with a hypothesis, an approach to test the hypothesis, results, and a conclusion.

Experimental design

The research is within the scope of PeerJ.

The objective of the study is to identify, among peripheral blood T cells, markers of immunosuppressed and yet stable immune state versus normal. It is unclear why this is important.

The experimental design has a significant flaw of not controlling for a potential confounder. It is known that renal failure and dialysis change patient's immune profile (Betjes, Michiel GH. "Immune cell dysfunction and inflammation in end-stage renal disease." Nature Reviews Nephrology 9.5 (2013): 255.). This work compares immune profile of patients with previous renal failure with normal subjects. Although renal function in the enrolled subjects have been corrected with the transplant, it is unclear if transplant also revert immune dysregulation. Thus, differences the authors find may be due to previous renal disease or immunosuppression. The authors did not consider the impact of previous renal disease and attributed differences only to immunosuppression. One way to correct this is to enroll patients with renal failure before transplant not on immunosuppression to see if the observation is unique to stable immunosuppressed patients.

The technical aspect of this work is strong. There are reasonable number of patients in each group (44 to 52). The flow panels are rationally designed. The flow cytometry profiles look very robust. Statistical analyses are appropriate. The study has minimal risk to patients and it is conducted after obtaining informed consent.

Method is mostly sufficient. Some clarification needed:
1. How many subjects for each category (Y1 and Y5 after transplant) are screened to identify the 51 and 52 subject who meet the eligibility criteria? This may help to understand how selective and representative the study enrollees are.
2. Are patients taking immunosuppressive drugs other than tacrolimus? Typically renal transplant patients are on three or least two immunosuppressive drugs that include a CNI, MMF with or without steroid.
3. Were the patients on dialysis before transplant? This is relevant because dialysis can impact immune profile.

Validity of the findings

As stated above, I think this study lacks a critical control group to make the conclusion that the authors are making.

Otherwise, the data and statistical methods are sound. Presentation of the data is objective, clear, and focused on the research question.

Additional comments

The informed consent is in English. Is this a translation or all patients in the study are able to read and understand English?

---

## Round 0.2 · Minor Revisions

I confirm that the manuscript is scientifically acceptable but the grammar and English language still needs work. I understand that you contracted with Springer but they have not done a good enough job editing. Please get a fluent English speaker to edit it further.

---

## Round 0.3 · accepted · Accept

Authors and production staff should confirm that the English is satisfactory

#